# Digital Animism: Towards a New Materialism

**Victor J. Krebs**

Humanities Department, Pontifical Catholic University of Peru, Lima 15073, Peru; vkrebs@pucp.pe

**Abstract:** With the advent of 'the virtual world,' we have naturally gauged the 'reality' of the virtual in terms of how close it comes to empirical experience. However, the common association of the virtual to simulation depends on a representational dualism that reduces it to a simulacrum of reality and prevents us from seeing its real import. Virtuality, rather than related to simulation, refers instead to potentiality. Far from being something that first appears with the digital-virtual as a technological simulation, the virtual constitutes the bare potentiality intrinsic to human experience, always subject to technological modulation. Despite the path of increasing abstraction marked by the evolution of the technologies of communication, I argue that the virtual world, paradoxically, reveals matter as ineluctably vital and in permanent movement and transformation. The digital thus does away with the dualism responsible for the modern disenchantment of nature and—decentering the human, placing it as equally part of a rhizomatic and entangled nature—lays the groundwork for an animistic ontology that is consonant with a new materialism.

**Keywords:** animism; natural religions; philosophy of technology; new materialism; virtuality; digital life

## 1. Introduction

Matter disenchanted.

For the Western mind, at least since the XVIIth Century, matter is inert, solid, measurable (has length, breadth, width), is quantifiable, subject to cause and effect and ontologically other to Descartes' *res cogitans*, which is, on the contrary, agentic, immaterial, and capable of manipulating and reconfiguring matter. Our modern attitude of domination and control over nature, as well as our disregard for the life of nonhuman animals, is grounded on that dualistic vision.

On the other hand, the history of technology has arguably been defined by the (not unrelated) systematic, and instinctive disavowal of mortality. It has resulted in a process—which Flusser (2011) describes in his analysis of the development of the technology of communication—of fleeing the concrete in an increasing abstraction. The human subject has tended to disengage from sensible experience, advancing towards ever more schematic ways of configuring and understanding reality. From the four-dimensionality of our precognitive immersion in nature, Flusser claims, we have advanced to the three-dimensionality of action in the world, then to the two-dimensionality of its representation (in images drawn and painted on flat surfaces), and, subsequently, to the one-dimensionality of sequential lines of graphemes on paper to signify it until, finally, in the 21st Century, we have arrived at the zero-dimensionality of the digital. At this highest degree of abstraction, the world has become a digital construction of thought and no longer an empirical representation of what we perceive.

That zero-dimensionality attained by digital technology has brought about the advent of virtual reality and is radically transforming the way we apprehend and inhabit the world. This zero-dimensionality of the digital does not involve a denial of the dependence of digital technology on material supports, but rather refers to the modulation of reality the digital effects that abstracts the data of perception into the punctual dimension of the numerical code. The digital and the virtual seem to provide us a new horizon and a

silent promise of overcoming our limitations, of transcending our mortal condition. We are therefore (wrongly) persuaded to think of the virtual as a second "world", an artificially constructed environment that emulates, and sometimes amazingly mimics, reality. We (mistakenly) extend Cartesian dualism to our now (also) digital life and, by placing the hope of freedom from matter in the virtual, we perpetuate the estrangement from matter it enabled in the first place. However, the digital may be, paradoxically, providing us the tools to overcome that estrangement, reconceive matter, and subvert the Enlightened secular stance of modernity towards the world.

In what follows, I want to argue that if we liberate our understanding of the virtual from its dualistic framework and consider what the digital modulation is offering us in terms of understanding the world and our relation to it, a new way of conceiving matter is made available, where we can start talking of what I call a form of digital animism. By this I mean a view of matter where the agentic capacities of nature may become visible and thus operative for us again, without the anthropocentrism that incurs in the so-called pathetic fallacy that has traditionally neutralized in modernity the conception of matter as animated.

## 2. Against Dualism

### 2.1. The Virtual as Potency

Within the modern dualistic framework, we have naturally judged the 'reality' of the virtual according to how close it comes to empirical experience. But in that representational dualism, where the virtual simulates or provides a simulacrum of reality, we typically find ourselves split in two polarized stances. According to the first, the virtual—in the term "virtual reality"—stands for something that is fake, unreal, even illusory. At best, virtual objects in cyberspace are mere copies that only represent or replicate reality but will themselves never be quite real. A virtual hug will always be less than a real hug, a virtual tree less than an actual tree, and certainly a virtual friend nothing the same as a friend. According to the second polarized stance, however, we consider the virtual as an improvement, an upgrade of reality, rather than as a false copy. In this second perspective, technology can redeem us from a fallen world. Both stances, however, depend on the dualistic framework. For both the real and the virtual are "immutable and inalienable forms" (Doel and Clarke 1999, p. 270), each separate and protected from the 'degrading' effects of the other.

The actual meaning of virtuality, however, has more to do with force than with simulation. "Virtual" is derived from the Latin word virtus, which means strength or potency, so that "potentiality" is a more primary meaning. The virtual in this sense refers not to a simulation of reality, but as Frankel and Krebs (2022) have put it, to the "*dimension of possibility* inherent to all reality" (p. 4). As Brian Massumi puts it,

> [The virtual] concerns the potency in what is, by virtue of which it really comes to be. It connotes a force of existence: the press of the next, coming to pass. The virtual pertains to the power to be, pressing, passing, eventuating into ever new forms, in a cavalcade of emergence. (Massumi 2014, p. 55)

Further, it is our ability to "lift" the empirical world into the abstract dimension of concepts and images—the human gift of the tongue to articulate experience in words—that marks the emergence of what we might call the human virtual: a mental space wherein we are able to think, imagine, fantasize, and invent the world out of that atemporal and unchanging dimension of possibility that our words (and then every technology they make possible) open up for us. In that sense, language modulates the potency of things in the forms which our concepts give to experience. In some sense concepts are, as Niels Bohr claimed, "specific material arrangements", that modulate reality in what Karen Barad characterizes as "the materiality of discursive practices" (Barad 2012, p. 11). All other techniques—drawing or writing, photographing, filming, digitizing, etc.—extend our ability to give form to, and materialize, the potency in the world we experience. Each modulates virtuality in the various forms in which reality will be articulated for us, and

so constituted as conscious impressions and articulated experience; the digital virtual is merely the latest technological modulation of human virtuality.

Something important follows from all this, for far from being a copy of reality, the virtual must be considered, in Deleuze's famous words, "fully real in so far as it is virtual" and "[r]eal without being actual . . . " (Deleuze 1994, p. 208). Virtual and real are two aspects of a constant inter-flowing phenomenon, neither of them more originary or more authentic, neither competing nor threatening one another, for far from standing outside the real, the virtual is at the core of the real. The 'real' is not only the actual, of which the virtual is supposed to be a simulation, but also and at the same time virtual, for the virtual names the potentiality at the root of the real. Dualism is gone.

Now, if the virtual informs the real even before it is actualized, then everything is potentially changing always. The real is permanently pregnant with the infinite virtual. So, splitting what is *virtually* there from what is *actually* there, taking what is actual as the real and refusing to see the virtual as part of the real, betrays a desire to sediment the real by disavowing the potential at the root of all appearance, and resisting the change which it ushers in. Contrary to 'the dogmatic image of thought' (Deleuze 1983, p. 103) that weds reality to permanence and immutability, we are envisioning, with Deleuze's take on the virtual, an ontology of continuous transformation. Reality is an inexorable flow of potential and actual, that opens room for the acknowledgement of temporal movement and mutability, setting the stage for a new understanding of matter.

### 2.2. Immanent Virtuality

Thinking of virtuality outside of the underlying dualism, where it is hypostatized and conceived as a product in competition with actual reality, liberates it from the constraints of representation. As Doel and Clarke observe, the virtual outside of that dualism is "without original archetype or prototype" (Doel and Clarke 1999, p. 282). Rather than a reflection of something else, it is a sui generis interaction of pre-existing horizontal vital intensities, vectors, and lines of force, always formlessly active behind the sedimented identities. Or, perhaps we should say instead that it involves their "intra-action" given that, as Barad says,

> in contrast to the usual "interaction," which assumes that there are separate individual agencies that precede their interaction, the notion of intra-action recognizes that distinct agencies do not precede, but rather emerge through, their intra-action. (Barad 2007, Loc. 842)

Just as psychoanalysis has taught us to look at dreams, the virtual world teaches us to look at its simulacra (and hence at every modulation of reality) beyond their apparent forms. Despite our first impulse to equate, for example, the image of my brother in a dream with my brother, in working with the dream it soon becomes clear that this outward appearance is hiding a difference. The image in the dream has an autonomy of its own that breaks from its referential origin as a spontaneous product of the imagination. The virtual, too, Massumi writes, "bears only an external and deceptive resemblance to a putative model . . . [that] envelops an essential difference" (Massumi 1987, p. 91). If it "represents" it does so not just as another one of the same but as "a wholly transformative production of something other than the same" (Doel and Clarke 1999, p. 266). Its agenda is other than to copy a sedimented original, and its identity is autonomously forged, independent of any pregiven model. As Massumi explains,

> The thrust of the process is not to become an equivalent of the "model" but to turn against it and its world in order to open a new space for the simulacrum's own mad proliferation [where] it affirms its own difference. (Massumi 1987, p. 91)

What the virtual discloses is familiar to us not merely in dreams, but in the very operations of memory, where what we remember is always affected by the experiences that we have had in the interim after the event we remember, and the present circumstances that are themselves traversed by so many other lines of force in their unstoppable movement. Memories are never static units but "a flock of differentials" permanently touched by what

we perceive and experience, and so transformed into what they are now, where "now" is an indexical, always referring to this present moment and its current unceasing mutations.

It is important to note at this point, that behind our stable perceptions there is the ever-shifting vital multiplicity of the sensible. It is from the constellations of different intensities that freely circulate under our awareness that clear perceptions will eventually emerge into consciousness. What we normally perceive are in fact perceptions that result, as Andrew Murphie writes, from "a differential operation at the threshold of perception" (Murphie 2002, p. 199) that constellates our formless or "fuzzy perceptions" into something visible. It is important to note a caveat here: I am aware of Karen Barad's criticism of the analogies between 'macro' and 'micro' worlds underlying Murphie's claim as "flat-footed" (Barad 2010, p. 18) because they presume a given spatial scale. However, one need not take the difference here spatially but functionally. At the threshold of perception there is undifferentiation and a constant entanglement that produces out of the fuzziness or formlessness of the flux a sudden constellation that emerges into visibility from unconsciousness.

An image is formed, therefore, whenever a new significance arises from within the prior constellations that have become sedimented meanings in our habitual perception. It emerges out of the dance of invisibilities and sudden significances, differences, and relations at the threshold of awareness that constellate to create our clear perceptions. However, the image, conceived in this way, is not the static object that we perceive, somehow reflecting or duplicating something else. The image is not a re-presentation of something other, it is an emergence, an operation, an animated field, where what was once invisible breaks into visibility, where the unuttered becomes an utterance.

> The meaning and reality of the [image] is determined from within its own occurrence and not by any external model or criterion. [ . . . ] Differentials weave opposites together into a rhizomatic psychic and physical space, and the images produced [ . . . ] resemble nothing other than themselves in their own movement. (Frankel and Krebs 2022, p. 100)

## 3. The Field

### 3.1. Emergent Images

According to Greek mythology, all images originate in Hades, and more specifically in its darkest center: Tartarus, son of Ether and Gaia. They are all thus engendered from what is most ethereal and from what is earthliest, the most immaterial and the most material (Cf. Hillman 1979); they come from an ethereal dark and formless vitality. The appeal to the Greek myth here is an attempt to bring into our reflection an imaginal dimension which proves useful, as when considering the virtual we need to engage with the realm of the unsayable (or the negative, Bion's 'O', the unrepresentable, etc.). I believe this appeal is methodologically warranted if we want to be able to move beyond our modern framework and its constraints.

> What the human hand paints, draws, sculpts, writes, types; what the human figure traces with the infinite expressiveness of the face and the body; the material forms of nature and empirical reality, and even what the imagination conjures up: all physical and virtual images come from those depths. This object in front of me, our bodies, the skies above us, the aroma of the coffee that captures my senses, the beauty of the semblances that delight us on our smartphones, the voices that conjure up all sorts of feelings, moods, and emotions in you, etc., they are all images, and they all emerge from that abyss. The sensible images of perception, in other words, are as oneiric as are the images of our dreams and fantasies. Furthermore, as Rancière (2019, p. 7) reminds us, images are not merely perceptual (visual, auditory, olfactory, etc.). Images can consist wholly in words; they are all operations whereby we are constantly making the world visible by giving form to what is significant to us, bridging and integrating perception and affection, making the invisible visible. (Krebs forthcoming)

According to Carl Jung ([1960] 1981) it is naïve to assume that there is an identity between an image in internal reality and an object in external reality (par 516, p. 270). It is equally naïve to assume an identity between the external image and the object of which it is an image. For, as we are thinking of the image here, it may be considered a transitional object that hovers between consciousness and the unconscious, firmly holding the empirical and the subjective together occasioning a perception, bringing to visibility the virtual formlessness from which it emerges.

Images are autonomous operations that constitute what we experience, constellations of the possible that are nowhere governed by any model. Their appearance depends on a myriad of factors beyond what we are aware of, ranging from psychological (personal idiosyncrasies and inclinations, temperamental and cultural singularities), to material (geographies, histories, climates, habitats), and technological. Ultimately, there is an underlying spontaneity that far transcends the subject's consciousness occasioning their emergence.

However, there is an implicit commitment to permanent and stable essences or substances in the prevalent conception of thought as representational in modern culture, that desensitizes us to the constant flux behind what we take as stable entities and fixed states of affairs. Although, as Andrew Murphie points out, what virtual reality reveals is that the "simple facticity of stable bodies and fixed states of affairs" are simply "regimes of separation" (Murphie 2002, p. 192). They sediment life to make becoming more bearable, and respond to our need for control and domination of what is other and passing. For further elaboration on this point, see Krebs (2004, 2013).

### 3.2. Image Makers

With the analog image—say a photograph of someone—we know that the light that emanates from the paper and touches my eyes really came from the luminous presence that was imprinted in the photographic plate at the very moment when the person was photographed. However, the digital image, as Bernard Stiegler says, "breaks the 'umbilical cord'" (Derrida and Stiegler 2002, p. 152) that grounds the materiality of the process that generates the photograph. The digital image of someone is no longer tied physically to that person. There is no past, in other words, from which this image comes. This is Stiegler:

> With the digital photo, this light, from out of the night [ . . . ] doesn't come from a past day that would simply have become night (like photons emanating from a past object). It comes from Hades, from the realm of the dead, from underground: it is an electric light, set free by materials from deep within the belly of the earth. An electronic, decomposed light. (Derrida and Stiegler 2002, p. 153)

The digital image is not an actual footprint of anything; it is "an algorithmic phantom of something that may have never been" (Frankel and Krebs 2022, p. 36). It offers us an assemblage of computer data that may even be permuted and interlaced in ways that need not hold any relation to actual reality, but that actually can enter into our world and inspire our creativity or activate our complexes or even foist our terrors.

Flusser (2011) makes a related point when he observes that the traditional or non-technical image emerges from a world of objects, from which the image maker is directed to an actual surface. However, the situation is inverted in the digital case, where the image maker is "directed from a particle toward a surface that can never be achieved"; whereas the maker of traditional images abstracts and "retreats from the concrete", the digital image maker seeks "to turn from extreme abstraction back into the imaginable", "to make concrete". As Flusser sums it up:

> We are concerned here with two image surfaces that are conceived completely differently, opposed to one another, even though they appear to blend together . . . The meaning of technical images is to be sought in a place other than that of traditional images. (Flusser 2011, p. 21)

"Disconnected from space, the digital image is indeed unmoored from all familiar ports" (Frankel and Krebs 2022, p. 36). It is cut loose from the causal logic of reality as we have come to know it. By reducing the world to infinitesimal particles, to each of which we associate a sequence of 0 s and 1 s we have made a second, parallel world, that is weightless and indestructible, one we can store, transfer, and clone indefinitely and anywhere. We reproduce the image of my body, for example, from a mathematical grid, turning it into an algorithm from which my digital image originates, anywhere and anytime. The laws of empirical space belonging to my physical image no longer apply to my virtual self. The modifications of my face, for example, made possible by the multiple apps always trending on social media, liberate my image from the constraints of representation, generating a freedom of associations and connections that turns reality, our digital life itself, into a dreamscape.

We are now able to produce artificial images that, instead of representing the world, create illusions from which we construct the new realities that we then start to share and inhabit together. Alternative worlds start to emerge in the synthesized images on the screen: "lines composed out of point-elements, surfaces, soon also bodies and movable bodies", Flusser writes, "these worlds are colored and can sound, in the near future they will probably also be touchable, smellable and tastable" (Flusser 2002, p. 202). In fact, they already are in many of the videogames that feed the imagination and occasion the major involvement of a large portion of native digitals, and who knows what the Metaverse will bring forth.

What we are experiencing increasingly nowadays, not only with virtual reality but with that merger of digital virtuality and empirical experience that we can call our digital life, is that "mathematical thinking brings forth alternative worlds that freely begin to mingle with what was previously understood as reality" (Ieven 2003). We are, indeed, living in a time where digital entities enter our world (holograms, avatars, memes, gifs, etc.) not as copies that downgrade or upgrade reality, but as virtual simulacra (i.e., as spontaneous and creative emergences). They are active from within themselves and not in terms of any external or transcendent original.

### 3.3. Modulations of Reality

Given the impact our technological gadgets are having on how we experience the world, we begin to perceive and experience no longer in terms of substances, but in terms of processes. In the zero-dimensionality of the digital, lines have been transformed into networks, hierarchies into a single plane of immanence, sequences and chronologies into rhizomes and synchronicities. From considering things in terms of quantities we begin to see them as qualities, or as Deleuze (1993, p. 19) says, objects become 'objectiles', moving entities, events that disclose the simulacrum's grounding, "no longer in some ideal atemporal realm, but in an immanent world of temporality, a flock of differentials" (Frankel and Krebs 2022, p. 97).

In the move from substances to processes, from objects to objectiles that the virtual world introduces, a new logical framework becomes necessary. Whereas the guiding metaphor for the traditional logic that deals with substances was conceived in terms of the hierarchical, arboreous, monothematic kind of knowledge, in this new logic, it is rather conceived in terms of a horizontal, rhizomatic, pluralistic knowledge that is characterized more by circulations than by lines and angles:

> [U]nlike trees or their roots, [the rhizome] connects any point to any other point, and its traits are not necessarily linked to traits of the same nature; [ . . . ] It is composed not of units but of dimensions, or rather directions in motion. It has neither beginning nor end, but always a middle (milieu) from which it grows and which it overspills. (Deleuze and Guattari 1987, p. 21)

The constitution of our world, then, is piloted more by the associative powers of the imagination than the logical connections of the intellect. The rhizomatic deconstructs the scribal order, demolishes its hierarchies, bends its linearity and makes the idea of

representation, of there being a single referent to which things must assimilate, idle. The inherence of a system of referentiality that binds the simulacrum-as-copy to an original is abolished in such a logic. So is the dualistic framework that divides the virtual and the real. It explodes the sedimented and static model that provided fuel to the original-copy relation, diffusing a multiplicity of forces, directions of possible links, and associations that result in wholly new self-originating, autochthonous meanings in fluid horizontal interaction.

As Frankel and Krebs put it, "in the digital age we awaken to a new ecology of the virtual, where the actual is not only permanently open to its unpredictable multiplicity, but also inextricable from it" (Frankel and Krebs 2022, p. 101). The virtual simulacrum is not a mimesis but a 'modulation' that temporalizes experience and thus liberates it from the paralyzing expectation of stability. It is not that virtuality itself has been increased with the digital. Murphie (2002) points out that it is rather that our ability to modulate it has been dramatically enhanced. With our new technologies we can determine perceptually, affectively, even ontologically how the world will be open to us. By zooming in or out, fast-forwarding or rewinding, cropping or reframing our images we can intensify or simply eliminate aspects of the real, thus reconstituting our experience. The digital provides tools that give us more power than ever for modulating that permeability of the real—reassembling its original order, merging different temporalities to our own experience of time—making visible dimensions of the virtual that had remained invisible until now.

> In replaying in slow motion, the capture of our peeling an orange, for instance, the pressure of our fingers against the peel is evoked in the experience of the screen images, where the conjunction of our bodily memory and the audiovisual feed unveils aspects of our subjective experience that usually go unnoticed. The fleshiness of the fruit, the precise movements of our fingers in interaction with the orange, the feelings, sensations, and associations that it calls up and so on, all aspects which would otherwise have remained buried in the darkness of unconscious oblivion now open into consciousness, providing in this encounter, the rudiments of new vocabularies and common discourses that can broaden the range of our lived experience and broaden the confines of our world. (Frankel and Krebs 2022, p. 102)

Radical changes are occurring in the virtual world. The rhizomatic expansion and diffusion of experience that we are witnessing in the digital conspires in its openness and ontological promiscuity against the rationalistic ideal of closure and completeness to which we cling as if it were indispensable for our very survival. We are able now to begin to see the world beyond the sedimented perceptions that had constituted our familiar pre-digital world, and to respond to their obstinate rigidity, purposely affect it by modulating what had previously remained unconscious or disavowed, materializing its free virtuality.

Of course, technology is a pharmakon, it is at the same time a poison and a remedy, depending on the dosage with which we apply it, and it certainly carries with it the peril of deepening the sedimenting and sedimented mentality of the modern scientific/neo-liberal/capitalist/colonialist mindset, enslaving us and stupefying us, as Bernard Stiegler never tired of warning us. However, one of its potentials lies in its instilling in us a heightened sensitivity and responsiveness, in perception and action, to an environment that is always in flux, never the same from one moment to the next, traversed by time and temporality, and a vitality that flows through all. It opens our eyes to a world ensouled, psychically active beyond the egocentric human self.

## 4. Conclusions

> *Matter and meaning are not separate elements that intersect now and again. They are inextricably fused together, and no event, no matter how energetic, can tear them asunder.*
> —Karen Barad

Going back to where we started with the Enlightenment's disenfranchisement of matter, we find in this account of virtuality what could be seen as a route of reinversion

and reinvestment of matter, in an ontology of forces, energies, and intensities (rather than sedimented objects) and complex, even random, processes (rather than simple, predictable states) that displaces the substantialist Cartesian or mechanistic Newtonian accounts of matter. The dualistic perspective is abandoned in favor of a "a monistic vision of emergent, generative materiality", where the virtual is an immanent force or potency that underwrites all actualized and actualizable forms, as it were, an open-ended power of reorganization of what there is (cf. Bennett 2010, p. 57), an infinitely flowing reserve of formless vitality, a "pressing multitude of incipiencies and tendencies" in Brian Massumi's (1987) words.

However, this is all in tune with new materialist ontologies that are abandoning the terminology of matter as an inert passive substance subject to predictable causal forces. The modern mindset—anthropocentric and rationally imperialistic—disenchants matter and alienates us from a world where everything is seen as animated by a common principle. When I speak of a 'digital animism' I mean to point to the way in which the digital virtual re-opens that space, where the agentic capacities of nature may become visible and thus operative for us again. As Diana Coole and Samantha Frost put it in the introduction to their book *New Materialisms*,

> an excess, force, vitality, relationality, or difference [ . . . ] renders matter active, self-creative, productive, unpredictable. New materialists are rediscovering a materiality that materializes, evincing immanent modes of self-transformation that compel us to think of causation in far more complex terms; to recognize that phenomena are caught in a multitude of interlocking systems and forces and to consider anew the location and nature of capacities for agency. (Coole and Frost 2010, pp. 9–10)

Jane Bennett, in her book *Vibrant Matter*, for instance, tries to vindicate what she characterizes as "the negative power or recalcitrance of things. . . . the active role of nonhuman materials in public life". She also subscribes to W.J.T. Mitchell's distinction between objects and things, where

> objects are the way things appear to a subject—that is, with a name, an identity, gestalt or stereotypical template . . . [but] Things [ . . . ] [signal] the moment when the object becomes the Other, when the sardine can looks back, when the mute idol speaks, when the subject experiences the object as uncanny and feels the need for [ . . . ] a metaphysics of that never objectifiable depth from which objects rise up toward our superficial knowledge. (Bennett 2010, p. 2)

The sharp distinction there is between mechanical inorganic matter and organic systems in the modern perspective is no longer adopted by new materialism, where everything seems to be instead enmeshed in a network of intensities that may at any point acquire agency and effective force. Bennett refers to this stance as an "enchanted materialism," where agency is ascribed to inorganic phenomena. Trash, food, even the electricity grid all enjoy an efficacy that defies and does without human will. Further, in Karen Barad's agential realism, reality is not something substantialized and fixed or demarcated, and matter is always already entangled with discourse and action and other material processes in "an intra-active inseparability and inseparably enacting practices" of constituting phenomena.

It strikes me, then, that new materialists are indeed suggesting something coincident or at least compatible with what Tim Ingold identifies as an animic or animistic mindset. As he explains:

> It is within such a tangle of interlaced trails, continually ravelling here and unravelling there, that beings grow or 'issue forth' along the lines of their relationships. (Ingold 2003, pp. 305–6)

This tangle, he adds later, "is the texture of the world. In threading each thing its own path through the meshwork—they contribute to its ever-evolving weave." (Ingold 2006, pp. 9–20).

It all seems to add up to a new sort of vitalism, or an animism that cannot be accused, however, of resulting from the so-called pathetic fallacy. For, whereas in the conventional sense, in Coole's and Frost's words,

> agents are exclusively humans who possess the cognitive abilities, intentionality, and freedom to make autonomous decisions and the corollary presumption that humans have the right or ability to master nature [here, instead] the human species is being relocated within a natural environment whose material forces themselves manifest certain agentic capacities and in which the domain of unintended or unanticipated effects is considerably broadened. (Coole and Frost 2010, p. 10)

In this sense we are clearly moving into a posthuman conception of material agency "that limits humans' agentic efficacy" (Coole and Frost 2010, p. 14) and so discards the anthropocentrism characteristic of the humanist legacy, by making the embodied human component integral to the processes of materialization or actualization rather than transcendent agents acting from outside. Bennet, for instance, wants to "highlight the extent to which human being and thinghood overlap, the extent to which the us and the it slip-slide into each other [so that] we are also non-human and that things, too, are vital players in the world" (Bennett 2010, p. 4). The Promethean vanities of human mastery over nature are thus all banished.

Vitality here is like the virtual: immanent to the very process of the world's continual generation or coming-into-being. It is not a property of the world but more like a necessary condition for its generation. What we normally conceive as an already sedimented 'environment' can begin to seem, after our itinerary so far, more like a domain of entanglement, where it is the interactive or intra-active reality of the virtual and the actual that constitute what we may well call our digital life. In this contemporary "liquid" society, as Zygmunt Bauman (2000) has called it, the illusion of permanence is shattered by the speed and dialectics of the digital.

Emerson called "the evanescence and lubricity of all objects, which lets them slip through our fingers then when we clutch hardest" the most unhandsome part of our condition. As Stanley Cavell notes, the unhandsomeness here is our tendency to deny the standoffishness of objects precisely by clutching at them (Cavell 1989, p. 87). The ontological shift we have been describing, therefore, involves not just an epistemological but also an ethical conversion, for it can no longer be a matter here of relating to things by 'grasping' them through concepts, as if they were our guarantee of that possession. In other words, what needs to change is our defining our relation to the world as one of knowing through concepts and judgments, as if we could capture the essence of things through that single mode. Not 'grasping' but being open to the unknown and even the unknowable is what is necessary, but then also wonder and mystery, which had been exiled from the context of rational inquiry, must be recovered.

A change in attitude becomes necessary in this new domain of entanglement, an ethical turn, as it were, where thinking becomes a form of thanking and of praise rather than of domination and control. I have advanced this line of argument in Krebs (2022). We need to sensitize ourselves, as Bennett (2010) says, to "the impersonal life that surrounds and infuses us" and develop "a more subtle awareness of the complicated web of dissonant connections between bodies".

Going back to Mitchell's distinction between things and objects, we could agree with James Hillman, who claims we need to develop a new nose of common animal sense, an aesthetic response to the world that ties the individual soul immediately with the world soul (Hillman 1992, p. 105). As he explains:

> Thing-consciousness could extend the notion of self-consciousness from the constrictions to subjectivism. An analyst sitting in his chair all day long is more aware of the faintest flickers of arousal in the seat of his sexuality than of the massive discomfort in the same seat brought by the chair: its wrongly built back, its heat-retaining fabric, its resistant upholstery and formaldehyde glue. His animal sense has been trained

to notice only one set of proprioceptions to the exclusion of the psychic reality of the chair. A cat knows better. (Hillman 1992, p. 114)

The enlightened mind exiles the opacity of material vitality, thus dividing and imposing a hierarchy of importances, a dualism between the cognitive and the aesthetic that may be serving merely the need, as Karen Barad (2012) suggests, of safeguarding hegemonic power and normalcy against the chaos of multiplicity, movement and change. The monistic perspective of new materialism rather favors and celebrates our becoming infected with all kinds of queer Others.

In reconfiguring our understanding of matter, we open space for any plausible account of coexistence and its conditions in the twenty-first century for, as Coole and Frost also point out, "in this multiply tiered ontology, there is no definitive break between sentient and nonsentient entities or between material and spiritual phenomena" (Coole and Frost 2010, p. 10). What we are talking about, then, is a modified conception of our place in the world, a democratization of our material entanglement, and the demand for a more receptive mode of relating to everything that surrounds us in a posthuman world. Perhaps more, it is instilling in our relation to nature the astonishment and wonderment that was banished from the mindset of modernity and considered inimical to science, but which need not be so in this new century.

**Funding:** This research received no external funding.

**Institutional Review Board Statement:** Not applicable.

**Informed Consent Statement:** Not applicable.

**Data Availability Statement:** No new data were created or analyzed in this study. Data sharing is not applicable to this article.

**Conflicts of Interest:** The author declares no conflict of interest.

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
