# Peer review of "Digital Animism: Towards a New Materialism"

_religions, doi:10.3390/rel14020264_

Round 1
Reviewer 1 Report
This is an interesting paper, congratulations on your good work!
I have some remarks, which might be due to lack of expertise in some of your research areas – please excuse me for that.
1.
You speak of the “zero-dimensionality of the digital”. I do not understand how the digital is zero-dimensional. The physical processes (at least until today) are happening in a temporal sequence and our perception of the digital seems to happen (as our thoughts often seem to do) in a sequence. It seems to me that digital is not non-temporal. At least not in the Western lines of thought of Aristotelian and Kantian categories – which is, of course, not a tradition one has to adopt, however, the sources you cite seem to indicate you are thinking within these paradigms.
Similar problems arise when you talk about having made “a second, parallel world, that is weightless and indestructible, one we can store, transfer and clone indefinitely and anywhere.” Do you detach the digital completely from
a.) its material component (0 and 1 are impulse or non-impulse as a physical event)?,
b.) the consequences of the material component (e.g. that those components are not available for everyone or that environmental conditions can, in fact, destroy the digital)?
c.) it being constructed around our perceptive apparatus?,
And why is this a parallel world – and not a phenomenon amongst phenomena, a phenomenon in reality, a phenomenon in the world?
I apologize, perhaps these are all questions that have been answered in the literature, but it seems to me that the description of the digital is, to me, a bit too much of a praise.
2.
You writhe: “We extend Cartesian dualism to our now (also) digital life” This would require some clarifications. The Cartesian dualism thinks of res extensa and res cogitans, but what res is the digital now? You seem to think (see above) that it is not a rest extensa (being weightless etc.) – would it be then a res cogitans? What would be the “I” (the “sum”) of that res cogitans? If it is still the human “I” then the human “I” does not necessarily have to be a human being (or to speak in Cartesian thought, an angel). This, however, would no longer be a Cartesian thought. You are by no means obliged to adhere to Cartesianism, however, if you use this terminology you might need to explain the way you develop it beyond Descartes.
3.
You write: ““Virtual” is derived from the Latin word virtus, that means strength or potency, so that its primary meaning is “potentiality.””
This depends on the Latin you are referring to; the meaning of “potentiality” is significantly younger than the meaning of “excellence” – so it is difficult to claim that this is the primary meaning.
You go on to say: “The virtual in this sense refers not to a simulation of reality, but to the dimension of possibility inherent to all reality.”
There is a difference between possibility and potentiality. One could argue (following Barbara Vetter, Potentiality and Possibility) that potentialities are properties of individual objects, while possibilities are individuated in their manifestations alone (https://philpapers.org/rec/VETPAP). This would be a relevant difference in your case.
Furthermore, if you claim that “virtual in this sense refers not to a simulation of reality, but to the dimension of possibility inherent to all reality” would you not have to give the claim that the digital is a parallel world? Is is not, being a “possibility inherent to all reality” an actualized possibility within one and the same world?
4.
You write: “Now, if the virtual informs the real even before it is actualized, then everything is potentially changing always. The real is permanently pregnant with the infinite virtual.”
I would appreciate some further explanations on this. In your ontology the virtual/possible reality informs the factual reality. This means basically that only what is possible can be real and what is impossible cannot be real. But why you this mean that “everything is potentially changing always”? Where does the “changing always” come from?
I am not sure about the metaphor. The real is pregnant with the virtual – but does not the virtual precede the real? Is not an entity possible before it is actual? Would that not mean that the virtual is pregnant with the real?
5.
You write: “For the Greeks all images originated in Hades -specifically in its darkest center: Tartarus, son of Ether and Gaia- thus engendered from the most ethereal and the earthliest, from the most immaterial and the most material (Cf. Hillman 1979).”
This statement is too general; this requires references to a specific (Greek) author; this is certainly not true for Isocrates, for Antiphon or Democritus.
I will leave it to this, but I would have quite a few more things that confuse me, but, again, this might be lack of familiarity with the topic.
Reviewer 2 Report
Very good, interesting article; this makes a coherent argument in favour of a new concept of digital animism. it may be because it is outside may field but I did wonder if it would profit from a definition, early on in the article, of animism from which we have become alienated in our human-centred culture. It might widen the readership and understanding of the article (but as I say this might be my 'left-field' perspective). I don't like the use of hyphens when not separated from following text - but again this might be in the house styler so, if so, ignore me.I think the first paragraph of 2. immanent Virtuality is incorrectly inset - I don't think its a quote but the second paragraph is so it might have been block inset incorrectly.
There are so many great passages of insight and writing here I believe this could be valuable in inter- and trans-disciplinary areas.
